# Efficient Scheduling of Home Energy Management Controller (HEMC) Using Heuristic Optimization Techniques

**Zafar Mahmood** [1], **Benmao Cheng** [2,*], **Naveed Anwer Butt** [1], **Ghani Ur Rehman** [3], **Muhammad Zubair** [3], **Afzal Badshah** [4] **and Muhammad Aslam** [5,6]

1   Department of Computer Science, University of Gujrat, Gujrat 50700, Pakistan
2   Jiangsu Key Lab of IoT Application Technology, Wuxi Taihu University, Wuxi 214064, China
3   Department of Computer Science and Bioinformatics, Khushal Khan Khattak University, Karak 27000, Pakistan
4   Department of Computer Science & Software Engineering, International Islamic University, Islamabad 44000, Pakistan
5   School of Computing Engineering and Physical Sciences, University of West of Scotland, Paisley G72 0LH, UK
6   Scotland Academy, Wuxi Taihu University, Wuxi 214064, China
*   Correspondence: chengbm@wxu.edu.cn

**Abstract:** The main problem for both the utility companies and the end-used is to efficiently schedule the home appliances using energy management to optimize energy consumption. The microgrid, macro grid, and Smart Grid (SG) are state-of-the-art technology that is user and environment-friendly, reliable, flexible, and controllable. Both utility companies and end-users are interested in effectively utilizing different heuristic optimization techniques to address demand-supply management efficiently based on consumption patterns. Similarly, the end-user has a greater concern with the electricity bills, how to minimize electricity bills, and how to reduce the Peak to Average Ratio (PAR). The Home Energy Management Controller (HEMC) is integrated into the smart grid, by providing many benefits to the end-user as well to the utility. In this research paper, we design an efficient HEMC system by using different heuristic optimization techniques such as Genetic Algorithm (GA), Binary Particle Swarm Optimization (BPSO), and Wind Driven Optimization (WDO), to address the problem stated above. We consider a typical home, to have a large number of appliances and an on-site renewable energy generation and storage system. As a key contribution, here we focus on incentive-based programs such as Demand Response (DR) and Time of Use (ToU) pricing schemes which restrict the end-user energy consumption during peak demands. From the results figures, it is clear that our HEMC not only schedules all the appliances but also generates optimal patterns for energy consumption based on the ToU pricing scheme. As a secondary contribution, deploying an efficient ToU scheme benefits the end-user by paying minimum electricity bills, while considering user comfort, at the same time benefiting utilities by reducing the peak demand. From the graphs, it is clear that HEMC using GA shows better results than WDO and BPSO, in energy consumption and electricity cost, while BPSO is more prominent than WDO and GA by calculating PAR.

**Keywords:** optimization techniques; demand-supply system; energy consumption patterns; genetic algorithm; particle swarm optimization; wind driven optimization; home energy management controller

## 1. Introduction

The energy requirement of commercial as well as residential users is increasing day by day. At the same time, a different generating unit of electricity faces a shortage of energy due to line losses and unpredictable energy demand from the end-users. For this reason, engineers and scientists are looking to adopt and implement a strategy that must be safe with reliable transmission and delivery. The researchers in this field are also looking to establish interactive communication between the end-user and utility by introducing

advanced information and control technologies [1]. As a result of different efforts, they proposed and developed different solutions based on smart grid technology to improve the reliability of the grid, thus providing communication between the end-user and the utility by minimizing the environmental issues caused by fossil-fueled generation [2]. Around the globe, the energy requirements of end-users and industry are increasing linearly, while power generation from different sources and the reliable transmission of the power is much slower than the consumption [3]. The traditional grid lacks communication between the users and the utility causing inefficient operation on both the demand and supply sides.

Smart grid technology majorly relays on different types of renewable energy and efficient demand-supply management. With the increasing energy consumption demand, requirements to switch from traditional fossil-fueled generation to smart grid technology will become a prominent research area [4–6]. As a result of the smart grid, the resultant energy will be environment-friendly, cheaper, and easy-to-use on-site energy, by addressing the stability and irregular nature of Renewable Energy systems (RES) [7–10].

A survey was held in the United States to determine household electricity usage. Based on the survey results, different household appliances consumed almost 42% of residential energy [11]. Researchers in this field propose and design new prototypes and standards based on energy consumption patterns for the residential electricity market by coping with energy optimization [12,13]. For this purpose, they introduced and deployed new technologies such as on-site RES and grid power, an advanced metering system, controllable appliances, an energy storage system, and intercommunication between utility companies. For this purpose, they introduced and deployed new technologies such as distributed energy generation (on-site RES and grid power), an advanced metering system, controllable appliances, an energy storage system, intercommunication between utility companies and end-users, and a stand-alone storage system. Using the two-way communication mechanism, end-users will be permitted to access their energy usage and pricing information. On the other hand, introducing the different pricing schemes at the retailer level allows an opportunity for the electricity consumer to minimize his electricity bills by shifting from peak hour to shoulder [14]. As we do not have a large-scale energy storage system, a balanced mechanism for energy generation and consumption must be implemented to avoid complete shutter-down and load-shedding problems [15]. In this article, we propose a mathematical model and implement a controlling mechanism:

- The Home Energy Management Controller (HEMC) enhances energy efficiency and improves the comfort level within a single residential home, without taking into account the retailer side data for load forecasts and different pricing schemes for scheduling.
- Using a two-way communication mechanism a user can shift appliances from high to off-peak time by increasing high demand at a particular time interval.
- To achieve this shift from high to low-peak, another mechanism Demand Response (DR) is introduced which is in response to the changes in the price of electricity over a certain time interval offered by the utility company, end-users also change their electricity consumption patterns from their normal routine to achieve many benefits from the subsidized patterns. In this work, we proposed HEMC based on load shifting technique and user preferences.
- We consider a typical home with 10 different types of electrical appliances, having a variable length of operation time to show the effectiveness of the proposed algorithms.

For energy optimization problems different researchers proposed different techniques such as Linear Programming (LP), Dynamic Programming (DP), Artificial Intelligence (AI) inspired techniques such as Genetic Algorithm (GA), Binary Particle Swarm Optimization (BPSO), Ant Colony Optimization (ACO), Wind Driven Optimization (WDO), etc., for energy consumption patterns, minimizing electricity cost calculation, maximizing user comfort and PAR based on different pricing schemes, different types of appliances having different operation time. We solve the same problems using GA, BPSO, and WDO with a different load-shifting strategy to achieve better results. Our proposed algorithm considers

two types of energy, grid energy and RES, with the storage system. For grid energy, we use the ToU pricing scheme which is fixed for a duration or season.

A typical home with an energy management system is depicted in Figure 1. This figure also represents the flow of the paper and the system model deployed in the paper. The proposed system model consists of RES which may be Photo Voltaic Cell (PVC), wind energies, and power utilities that supply electricity from the main grid. Electricity from the power grid is directly transmitted to the smart meter, whereas renewable energy is first transmitted to the proposed HEMC and then stored in the storage system deployed in the smart home. All the schedulable appliances in a typical smart home are connected with the proposed HEMC system, which optimally scheduled their operation to switch them to renewable and power grid to save costs while mainlining the user comfort levels. On the other hand, non-schedulable appliances which are fixed to be operated or based on their demands (while maintaining user comfort levels) also directly communicate with the proposed HEMC system to switch their operation on the RES system or power grid to reduce the cost and PAR values.

In the future, more incentives will be offered by different retailer companies to fascinate the user by reducing the peak-to-average ratio [16–18]. Both customers and the utility companies take advantage by using a demand response program [19,20]; the utility companies introduce different pricing schemes based on energy consumption at a certain time unit, which encourages the customer to shift their requirement from peak demand in response to the subsidized incentive [21,22]. A typical home with an energy management system is depicted in Figure 1.

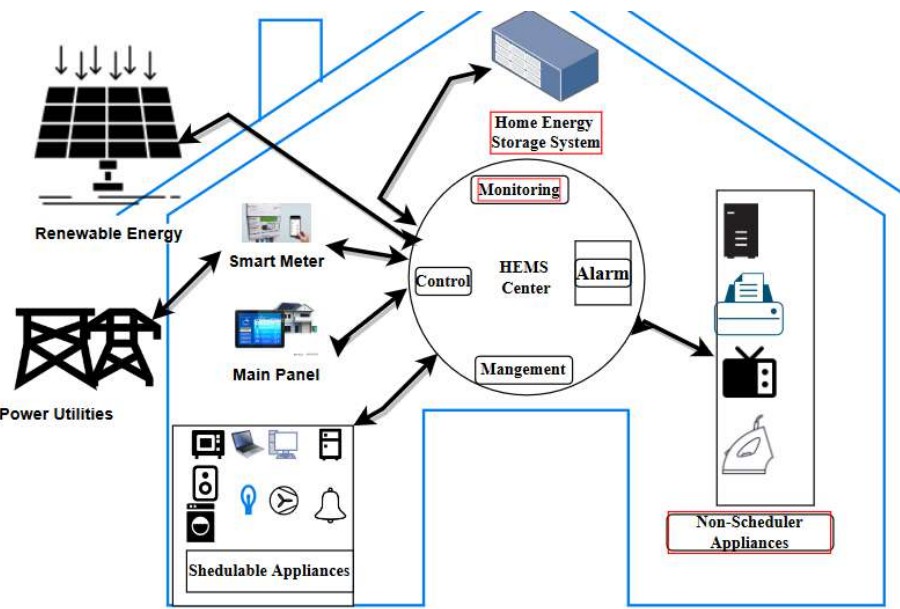

**Figure 1.** Overall structure of the proposed system.

Figure 1 also represents the flow of the paper and the system model deployed in the article. The proposed system model consists of RES which may be Photo Voltaic Cell (PVC), wind energies, and power utilities that supply electricity from the main grid. Electricity from the power grid is directly transmitted to the smart meter, whereas renewable energy is first transmitted to HEMC and then store in the storage system deployed in the smart home. All the schedulable appliances in a typical smart home are connected with the HEMC system, which optimally scheduled their operation to switch them to renewable and power grid to save costs while mainlining the user comfort level. On the other hand, non-schedulable appliances which are fixed to be operated or based on their demands (while maintaining user comfort levels) also directly communicate with the HEMC system to switch their operation on the RES system or power grid to reduce the cost and PAR values.

The rest of the paper material is structured as follows: Section 2 discusses the related work. Section 3 discusses the proposed schemes and system model architecture, and the appliance energy consumption patterns are discussed, respectively. In Section 4, we describe the different algorithms used in the smart scheduler, and a detailed discussion of the load optimization problem, WDO, and PSO algorithms is presented. The results of the simulations and performance evaluation metrics are given in Section 5, and Section 6 highlights the conclusion of the paper.

## 2. Literature Review

Energy consumption and optimization are the core issues in designing a smart grid while maintaining user comfort and reducing costs, as pointed out by [23–25]. These papers give an overview of PSO and its applications when used in different fields of life. The authors propose and design an optimization technique based on particle swarm optimization due to its robust nature and easy implementation in complex and non-linear problems. Due to its fast convergence nature, PSO-based scheduling offers an easy way to shift peak and high-peak demand to low-peak time intervals [26], at the same time allowing the end-user to use an incentive-based package to reduce the cost of electricity. This work rationalizes the importance and usefulness of DSM in efficient home energy management systems. To manage more expanded and elaborated ideas related to efficient energy management in the smart grid, such robust PSO optimization techniques can be used.

To cope with increasing energy demand effectively and efficiently, electric power supply and consumption patterns can be simulated for effective grid management. Over time, the generation capacity of electricity is decreasing day by day in traditional grids, facing several challenges related to delivering the increasing power demand. In traditional grids, data/information can only flow in one direction; from the grid to the end-used. To make two-way communication possible between the grid and the customer, smart grid technology is introduced, which intelligently interacts with the grid and user to efficiently carry and distribute viable, maintainable, cost-effective, cheap, and secure electricity supplies [27,28]. Following the smart grid mechanism, a scheduler is proposed and designed using different AI techniques to schedule different types of home appliances. AI-based techniques such as GA, PSO, ACO, WDO, etc., have been in use for the last couple of years for energy optimization problems. In the article [17], the author designed and implement a scheduling scheme based on PSO for different home appliances with a predefined length of operation time. The paper also presents a comprehensive review of PSO with its other counterpart algorithms. The proposed scheme takes both the interruptable and uninterruptible load for the scheduling. The authors in [29] integrate a fuzzy logic-based thermostat with the home energy management system to minimize battery degradation to prevent unnecessary renewable energy arbitrage. The proposed system employs day-ahead load scheduling to save costs, offers the best possible Demand Response (DR) and Photovoltaic (PV) self-consumption, and the fuzzy logic-based controller aims for effective DR of air conditioning while maintaining thermal comfort. Reinforcement learning is used by the authors in [30] to construct a home energy management system. The home electric appliance systems, which contribute to the most important loads in a household, are regulated by HEMC based on machine learning and reinforcement learning, allowing consumers to save power while still improving their comfort. By correctly optimizing and addressing the optimal use of renewable energy sources, the proposed method is examined for monitoring home electric appliances to reduce energy consumption.

Heuristic algorithms play an important role in energy optimization problems [31]. Paper [32] highlights a genetic algorithm-based smart scheduler for optimal power scheduling, and scheduled home usage appliances of different types in the HEM system. The author used time-of-use pricing signals and real-time pricing schemes for scheduling purposes, to minimize total energy cost and energy consumption. A comparison with unscheduled loads is carried out in the paper, which proves that GA shows more prominent results

in cost reduction and energy consumption. The DR program based on GA and PSO is proposed in [33], by sightseeing the appliance scheduling schemes to help the end-user ease by minimizing the electricity bill, without compromising his/her comfort. At the same time, the proposed DR program facilitates retailer companies to stabilize the grid by optimally reducing peak demand. The simulation was carried out for unscheduled and scheduled DR programs by implementing GA and PSO as a hybrid approach. The results show that the hybrid approach has better insight into energy consumption patterns as compared to BPSO, whereas in cost analysis, the hybrid approach shows more prominent results than unscheduled ones. A similar concept is also highlighted in [30].

The authors in [34,35] propose and implement a demand-side management scheme to reduce PAR and minimize the electricity cost while considering user preferences. The proposed scheme is based on a heuristic-based evolutionary algorithm by shifting the load during high-peak hours to facilitate both the customer and grid. Both papers deploy on-site energy generation and storage unit, which is integrated with the grid energy. The results demonstrate that during high-peak hours, some of the load is shifted to RES by reducing PAR and minimizing the cost. The technique in the papers scheduled a large number of appliances of different types. The authors in [36,37] use a heuristic-based evolutionary algorithm to reduce the PAR and minimize the cost by shifting the peak hour load to different sources.

The author in [38] studied a typical home load management problem for different classes of appliances using the ToU pricing scheme. Appliances are categorized based on the length of operation time and energy consumption. The author first proposed and developed an efficient mathematical model for different classes, using this model an efficient and optimal algorithm is designed to condense and decrease the overall electricity consumption and bill as well as peak lessening and saving during subsidized hours by maintaining user comfort levels. Time scheduling flexibility is introduced for each class of appliances so that users can adopt any model based on requirement and priority. Simulation results demonstrate that the proposed algorithm optimally scheduled the home appliances based on energy consumption requirements and patterns, the ToU pricing scheme, and operation time.

In [39], the author proposed a demand-side management technique using WDO and BPSO algorithms. The smart meter communicates with both the grid and the end-used, taking price signals directly from the grid and energy demand and requests from the different appliances. The energy management controller takes this information from the smart meter and performs its calculation to schedule all the appliances, by keeping in view the peak hour and price signal. The schedule was transmitted to the end-used's appliance and grid company. Furthermore, in the suggested model, the author tries to balance electricity cost and waiting time for different appliances to provide benefits not only to the utility company but also to the end-user. To further reduce and minimize the cost, a well-known mathematical problem formulation technique "min-max regret knapsack" is used, and this technique is compared to that of simple optimization algorithms. The simulation results show that WDO gives better results than BPSO for the waiting time of appliances and cost reduction.

The authors in [40] proposed an HEM system for residential users to reduce their electricity costs and PAR using two approaches, HEM with microgrid and HEM without microgrid. The proposed HEM not only scheduled the home appliances but also electric vehicle charging and discharging optimally while maintaining user comfort. Each end-user has its microgrid which is connected to their grid, having a solar panel, gas turbine, wind turbine, and energy storage system (ESS). The authors use linear programming techniques to formulate the scheduling problems. The simulated results demonstrate that linear programming techniques can efficiently schedule different smart appliances and electric vehicles according to electricity generated by the microgrid. Using microgrid generation causes a reduction of the PAR and total cost of electricity. The authors in [41] designed a mathematical model to integrate different energy sources having small-scale generation

capacity. The model, which is based on an intelligent multi-objective named home energy management (MOHEM), aims to reduce the end-user electricity bill along with the system's peak demand by efficiently scheduling a smart residential home. The authors use the super criterion approach and the Pareto optimal solution ideas to deploy the cooperative game theory approach.

To improve the resiliency of the system, the authors in [42] proposed a new approach that is based on a genetic algorithm to empower the system planners to effectively handle different resiliency matrices in a bi-objective optimization planning model. A novel mixed-integer model was proposed by the authors in [43] to control the performance and efficiency of a LESS when used in conjunction with a DR scheme. The suggested approach includes cutting-edge managerial options, such as different DR activities that are permitted and the quantity of charging and discharging that is permitted. Furthermore, the model is built to be able to filter out the times when the demand side is prohibited from engaging in DR. The authors in [44] designed a multi-objective-based model to efficiently operate the demand and supply of a Smart Microgrid (SMG). The basic aim of the proposed model is to minimize the operation cost of the model, discharging pollution chemicals, and customer desired demand and usage curve in the daytime. A major contribution of the proposed model is to introduce an objective function used by SMG operators to balance the customer demands according to the supply with shiftable loads. The author Finlay uses fuzzy logic and the weighted sum approaches to choose the best solution.

The authors in [45] proposed a model based on multi-objective optimization, using a hydrogen storage system (HSS) while considering responsive consumers (RC). The basis of the proposed objective function is to increase the reliability of the system and minimize the operational cost and the gap between the demand and supply of the electricity. The author further used the Monte Carlo simulation model to effectively deal with uncertainties in the system. The end model was deployed by using the Shuffled Frog Leaping Algorithm (SFLA), through which the non-dominated solution is generated. Fuzzy logic and the weighted sum approach were used for the best solution. The authors in [46] designed a technique called MORL (Multi-Objective Reinforcement Learning algorithm) to effectively deal with the demand response to reduce the energy usage pattern while maintaining user comfort. If the proposed scheme is compared with conventional approaches, the earlier scheme alleviates the result of different end-user preferences and handles the indecision of future prices and renewable energy generation. Table 1 presents the comparison of the existing schemes and the proposed scheme.

**Table 1.** Comparison of Existing Schemes and Proposed Scheme.

| Article with Authors | Limitations of Existing Schemes | Novelties of Proposed Scheme |
|---|---|---|
| Zhao et al. [32] | The authors used time-of-use pricing signals and real-time pricing schemes for scheduling purposes, to minimize total energy cost and energy consumption. | The proposed HEMC system is based on GA, BPSO, and WDO to reduce the PAR which will minimize the electricity bills and thus increase the user comfort level. |
| Aslam et al. [40] | HEM system based on linear programming is used for residential and electric vehicles, to reduce the electricity cost and optimize power consumption. | The proposed HEMC system uses an offline storage system (ESS), and a different heuristic algorithm to optimally schedule the home appliances to reduce the overall electricity bill designed for residential purposes. |
| Lokeshgupta et al. [41] | Multi-Objective Home Energy Management (MO-HEM) is proposed to handle a small home energy demand. The authors use the cooperative game theory approach in their study based on super-criterion and a Pareto optimal solution concept. | The proposed HEMC system is stimulated with a Smart Scheduler (SS) which act intelligently, and all the appliances use two communication mechanism to communicate with the utility companies and SS using a smart meter. |

**Table 1.** *Cont.*

| Article with Authors | Limitations of Existing Schemes | Novelties of Proposed Scheme |
| --- | --- | --- |
| Chen et al. [46] | Multi-Objective Reinforcement Learning algorithm (MORL) to design a DR program to minimize energy usage while maintaining user comfort. Energy consumption based on a 24-h equal interval pricing scheme is important for both end-user and utility companies, by distributing the load properly in the H-hour horizon to offer maximum offer in terms of electricity cost. | The proposed system uses a different heuristic algorithm based on a linear programming model to optimize the appliance's scheduling. |
| Bina et al. [38] | A typical home load management problem for different classes of appliances, using the ToU pricing scheme. Appliances are categorized based on the length of operation time and energy consumption. The author first proposed and developed an efficient mathematical model for different classes, using this model an efficient and optimal algorithm is designed to condense and decrease the overall electricity consumption and bill as well as peak lessening and saving during subsidized hours by maintaining user comfort levels. | The proposed HEMC implements a demand-side management technique, using WDO and BPSO algorithms. The smart meter communicates with both the grid and the end-user, taking price signals directly from the grid and energy demand and requests from the different appliances. The energy management controller takes this information from the smart meter and performs its calculation to schedule all the appliances, by keeping in view the peak hour and price signal. The schedule was transmitted to the end-user's appliance and grid company. |

## 3. Proposed Scheme and System Model for HEMC

This section carries the discussion of an ideal and ultimate approach for scheduling and managing the required power and power consumption of an ideal home having a large number of appliances is proposed based on a specific pricing scheme. Since most of the end-users still use traditional electromechanical meters, utility companies use Fixed Retailer Price (FRP) models for the end-user which is a fixed price all the time. Smart meters as a replacement for old and traditional methods will be used to record energy consumption reading in a real environment with high accuracy and minimum effort. These utility companies use different incentives and subsidy-based pricing schemes for the customer to reduce energy demand and thus stabilize the grid. Some of the important and more used pricing schemes are ToU and Real-Time Pricing (RTP): In the earlier pricing schemes, 24 h of the day are equally divided into equal intervals and the price for each interval is known in advance by the users. Thus, a user can schedule his/her appliances based on the price signals, i.e., in peak hours, the user tries to turn on fewer appliances to reduce energy consumption, and hence minimize the cost. While in the shoulder and off-peak interval, most appliances are turned on. The RTP is somehow like ToU, where the price is based on end-user energy demand varying each hour. In this work, we present an inventive prototypical strategy to determine the required energy and electricity usage pattern for typical home electrical appliances in advance. The proposed HEMC system is stimulated with a Smart Scheduler(SS) which acts intelligently, and all the appliances use two communication mechanisms to communicate with the utility companies and SS using a smart meter.

The smart meter receives a price signal from the grid and passes it to the SS, on the other hand, different appliances send on/off requests to a smart meter, which also passes the on to the SS. The SS knows in advance the operation time for each appliance. Taking all these inputs from the grid and appliances through the smart meter, the SS generates energy consumption patterns and schedules all the appliances in the given domain of search space according to the price signals and operation time to decide the optimal time for the smart appliances to minimize energy consumption, minimizing electricity cost, while considering user comfort and reducing the peak to average ratio. Our proposed schemes also consider on-site RES generation and storage systems. During off-peak time intervals when energy cost is minimal, the SS utilizes the grid energy, and when the grid energy cost

is at a maximum, the end-user shifts the load from the grid to the RES system to maximize the user comfort level. After performing many simulations, the results demonstrate the minimum cost in terms of electricity bills for the end-user having HEMC compared to those who have no infrastructure and installed architecture for HEMC at their homes.

Energy consumption based on a 24-h equal interval pricing scheme is important for both end-users and utility companies, by distributing the load properly in the $H$ hour horizon to offer the maximum in terms of electricity costs. Given a set of different home-based appliances, a great matter of concern is to carry and distribute the energy power load efficiently in the $H$ hour intervals, such that the installer of the HEMS obtains maximum profit out of the system, i.e., $A = a_1, a_2, a_3, \ldots\ldots\ldots\ldots, a_{24}$. Each listed appliance requires a different energy level to be operated and their consumption rating is shown in Table 6. Each appliance uses two-way communication with the SS of the HEMC. Smart grids and smart meters continuously exchange the demand and electricity cost frequently, as different utility companies offer different incentives and subsidy-based prices over 24-h time intervals, namely high-peak, low-peak, and off-peak, as listed in Table 7. The end-user tries to operate a maximum number of appliances in off-peak and low-peak hours, to fulfill his requirement, and operate fewer appliances in high-peak hours to minimize electricity bills, respectively. Reviewing the different pricing schemes and energy demand highlights that high-peak energy consumption will charge more to the customer, as compared to low-peak hours in a 24-h interval. Keeping in mind the different pricing schemes, each user optimally consumes energy, thus minimizing his/her electricity bill. We propose our model for the optimization problem given by Equations (1) and (2):

$$H = h_1, h_2, h_3, \ldots\ldots\ldots\ldots, h_{24} \tag{1}$$

$$A = a_1, a_2, a_3, \ldots\ldots\ldots\ldots, a_{24} \tag{2}$$

We divide 24 h into equal intervals, $H = h_1, h_2, h_3, \ldots\ldots\ldots\ldots, h_{24}$, in a fixed horizon of time interval h, the scheduling of every appliance must take into consideration different time bounds such as start time, finish time, and length of the operation time, $H_s$, $H_f$, and $H_{Iot}$, respectively. Each appliance may have a scheduling time interval between $[H_0, H_{max}]$, where each hour has a different price signal. During 24-h time intervals, the energy consumption vector is given in Equation (3):

$$E_T = E_1^{t_1}, E_2^{t_2}, E_3^{t_3}, \ldots\ldots\ldots\ldots E_n^{t_n} \tag{3}$$

where $E_1^{t_1}$ is the sum of consumed energy by the first appliance in a fixed time horizon $t_1$ and so on. The inclusive unbiased and objective function is to decrease the cost of electricity, formulated in Equations (4)–(6):

$$min(C_h) = \sum_{h_1=0}^{h_{24}} C_h \tag{4}$$

Subject to:

$$\sum_{a_1=0}^{a_n} C_h \sum_{h_1=0}^{h_{24}} C_h \left( E_{h_1} a_1 \right) \leq Egrid \tag{5}$$

where

$$1 \leq h_1 \leq h_{24} \tag{6}$$

where $h_1$ to $h_{24}$ represents the 24-h horizon from 0 to 24, $C_h$ represents the cost of energy at a particular hour, $a_1$ represents the set of the appliance, $E_{h_1} a_1$ represents the energy consumed

by the appliance $a_1$ during $h_1$ time horizon, *Egrid* and means energy from the grid. The end-user pays electricity costs in terms of electricity bills to the utility company for the energy consumption of different home-based appliances in a particular time interval over a 24-h horizon. The cost of appliances is the cost of energy consumption of a particular home-based appliance turned on in a specific time slot $h$. The cost is estimated mathematically by using Equation (7) and (8):

$$\sum_{a_1=0}^{a_n} C_h \sum_{h1=0}^{h_{24}} \left( E_{h,load}, *c_h \right) \tag{7}$$

$$\forall h \in h_1, h_2, h_3, \ldots\ldots\ldots\ldots, h_{24} \tag{8}$$

where $C_h$ is the electricity cost for the time interval $h$, $E_{h,load}$ is energy demand by the appliance a in a specific time slot $h$ and is calculated by using Equation (9):

$$\sum_{a_1=0}^{a_n} C_h \sum_{h_1=0}^{h_{24}} \left( E_{h,load}, *\alpha_{h,a} \right) \tag{9}$$

where $\alpha_{h,a}$ is a Boolean variable having the value 0 or 1, mathematically defined in Equation (10):

$$\alpha_{h,a} = \begin{cases} 1, & if(applianceisON) \\ 0, & if(ApplianceisOFF) \end{cases} \tag{10}$$

$\alpha_{h,a}$ represents the status of appliance $a$, the appliance operates and consumes energy in that specific time slot $h$ if $\alpha_{h,a}$ is 1, and off if $\alpha_{h,a}$ is 0. The smart meter receives an on or off signal from the SS, which then further communicates with all the household appliances and sends a control signal, i.e, $\alpha_{h,a}$ to different appliances to change their state. This is mathematically calculated by using Equation (11):

$$E_{h_1,a_1} = \begin{cases} E_a & if(\alpha_{h,a} = 0) \\ 0 & if(\alpha_{h,a} = 1) \end{cases} \tag{11}$$

where $h$ represents the time interval from 0 to 24 and presents appliance 1. In the proposed research work, we have $N$ number of household appliances in our home, so $\alpha_{h,a}$ is the $N$ binary bits pattern. As discussed in the previous sections, the HEMS are also equipped with renewable sources of energy to generate some part of the energy from photovoltaic plates. In our proposed model, we assume that at least 45% of its total energy demand will be generated by the RES and stored. Since RES cannot fulfill all the energy requirements of the end-user, the end-user must be connected to the main grid for the shortage of energy. Thus, the end-user will consume both the grid and on-site RES energy. The hourly energy production of a single photovoltaic module in $KW_h$ is given by Equation (12):

$$E_{RES,h} = \forall h\varepsilon(h_1, h_2, h_3, \ldots\ldots\ldots\ldots, h_{24}) \tag{12}$$

The RES generated energy added to the HEMC system from a non-site installed RES system is, therefore, using Equation (13):

$$E_{RES,h} = \sum_{a_1=0}^{a_n} \sum_{h_1=0}^{h_{24}} \left( E_{RES,h} \right) \tag{13}$$

The peak to the average ratio for GA, BPSO, and GA can be calculated as dividing the maximum energy consumption of all appliances by the average energy consumption in a particular time interval and is given by Equation (14):

$$PAR = \frac{max_{load}}{average_{load}} \tag{14}$$

In the proposed solution, we consider a typical home with $N$ number of appliances, with different power consumption rates and length of operation time. The energy consumption is calculated over a 24-h equal time interval. HEMC controls all household appliances by communicating with the utility which takes energy signals directly from the utility, and different appliances request through the smart meter. The scheduling time, i.e., a whole day is equally divided into slots. HEMC while considering the available energy capacity $C_t$ calculates the time-bound in terms of starting $T_s$ and finishing $T_f$ time intervals, as well as the energy consumption of each appliance in a given time interval. The energy consumption during all time intervals can be calculated by using Equation (15) and (16):

$$E_T = e_1^{t_1}, e_2^{t_2}, e_3^{t_3}, \ldots\ldots\ldots\ldots\ldots e_n^{t_n} \tag{15}$$

$$T = t_1, t_2, t_3, \ldots\ldots\ldots\ldots, t_{24} \tag{16}$$

The scheduling time horizon during which appliances can be scheduled is given by Equation (17):

$$T_{sch} = T_{max} - T_{lot} \tag{17}$$

where $T_{sch}$ is the time taken by SS to schedule an appliance, $T_{max}$ is the maximum time available for scheduling, and $T_{lot}$ represents the length of operation time. As WDO and BPSO have binary variables so particles are initialized randomly for binary positions as shown in Equation (18):

$$X_i \quad = \quad [X_i 1, X_i 2, X_i 3, \ldots\ldots\ldots X_i n], \forall X_i 1, X_i 2, X_i 3, \ldots\ldots\ldots X_i n \quad \in \quad 0, 1 \tag{18}$$

Each binary value having a probability of 0.5 is assigned to each particle in each dimension and is given by Equation (19):

$$Xi_d = f(X) = \begin{cases} 1 & if(rand \geqslant 0) \\ 0 & otherwise \end{cases} \tag{19}$$

where $d = 1\ldots\ldots\ldots, N$ represents the position of each particle in the $N$ dimension. To obtain the global best position the position of each particle is updated and is given by Equation (20):

$$X_{(i_d)}^{(k=1)} = f(X) = \begin{cases} 1, & if(rand \geqslant 0) \\ 0, & if(rand < sigmoid(V_{i_d}^{k=1})) \end{cases} \tag{20}$$

where the sigmoid function is calculated by using Equation (21):

$$sigmoid(V_{i_d}^{k=1}) = \frac{1}{1 + exp - (V_{i_d}^{k=1})} \tag{21}$$

After random initialization, each particle in the solution space moves randomly to avoid premature convergence, and the velocity $V_{(i,n)}$ of each particle is updated using the result given by Equation (22):

$$V_{(i,n)} = wv^t_{(i,n)} + \left(C_1 rand(1) * \left(p^t_{lb_{(i,n)}} - x^t_{(i,n)}\right)\right) +$$

$$\left(C_2 rand(1) * \left(p^t_{gb_{(i,n)}} - x^t_{(i,n)}\right)\right) \quad (22)$$

where $C1$ and $C2$ are the weights for the local best and global best of a particle moving with velocity y and position $x$. $rand(1)$ is a random variable whose value is between $[0,1]$, and $w$ is the inertia factor. The notations used in the proposed scheme are listed in Table 2.

**Table 2.** Summary of notations and symbols.

| Symbol | Description |
|---|---|
| $w$ | Inertia factor |
| $rand(1)$ | Random variable whose value is between $[0,1]$ |
| $C1$ | Weights for local best of a particle |
| $C2$ | Weights for global best of a particle |
| $V_{(i,n)}$ | Velocity of each particle |
| $T_{sch}$ | Time taken by SS to schedule an appliance |
| $T_{max}$ | Maximum time available for scheduling |
| $T_{lot}$ | Length of operation time |
| $C_t$ | Available energy capacity |
| $T_s$ | Calculates the time-bound in terms of starting and finishing time intervals |
| $T_f$ | Energy consumption of each appliance in a given time interval |
| $KW_h$ | Single photovoltaic module |
| $\alpha_{h,a}$ | Household appliances in our home |
| $h_1$ to $h_{24}$ | Represents the 24-h horizon from 0 to 24 |
| $C_h$ | Cost of energy at a particular hour |
| $a_1$ | Set of the appliance |
| $E_{h_1}a_1$ | Energy consumed by the appliance $a_1$ during $h_1$ time horizon |
| $Egrid$ | Energy from the grid |
| $E_T$ | Sum of consumed energy by the first appliance in a fixed time horizon $t_1$ |
| $H_s$ | Appliance tart time |
| $H_f$ | Appliance finish time |
| $H_{Iot}$ | Length of the operation time |
| $[H_0, H_{max}]$ | Appliance scheduling time interval |

## 4. Proposed Algorithm for HEMC

This section presents different algorithms for HEMC. The algorithm for WDO, BPSO, and GA is implemented in this section.

### 4.1. Genetic Algorithm

We propose and implement three different algorithms WDO, BPSO, and GA to manage the energy consumption and energy consumption patterns for a typical home that has a large number of appliances with different energy requirements and lengths of operation time. The resultant HEMC system not only generates an energy consumption pattern but also calculates the energy consumption and minimizes the electricity bill while considering user comfort and reducing PAR. To address all these problems, the design scheme should be able to tackle all these involutions. In the past, researchers have used different techniques such as LP and DP, but as the complexity of the problem increases, these techniques are not able to handle such a large number of appliances. Algorithms inspired by AI, such as WDO, GA, and BPSO have the potential to solve such types of complex problems. As compared to other algorithms, GA provides the finest solution for the cost optimization problem, for this reason, we use a GA-based scheduling algorithm. The smart meter interactively communicates with utility companies and appliances and sends the input to SS, using GA

for scheduling purposes. The HEMC controller cumulatively deals with the appliances in a defined time interval and gives a complete pattern by solving the minimization problem. The SS operates at the beginning of the day, after sending a request from the appliance to the SS controller, the action taken by SS is based on GA techniques used to schedule all the appliance's energy consumption patterns in advance. The chromosome configurations of GA represent the solution, i.e., a schedule for appliances of when and how to operate [47]. In this research work, the ON/OFF status of each appliance is represented by an array of bits. Thus, the length of chromosomes depends on the number of controllable appliances. Here in this work, we use 10 different appliances so:

$$Length\ of\ chromosomes\ N = (10) \tag{23}$$

where $N$ represents the total number of appliances. The initial population of chromosomes is randomly initialized, and the initial population is then sent to an objective function, which finds the fitness value for each chromosome.

The GA iterates the population many times, and in every iteration, as a result, a new population is produced by crossover and mutation. As we know that mutation rate and crossover directly affect the convergence of the algorithm, different techniques for crossover such as a uniform crossover, arithmetic crossover, two-point crossover, single-point crossover, and mutation can be used, and here in this work, we use single-point crossover and binary mutation. If we use a larger crossover rate, the algorithm will converge fast, and if using a larger mutation rate, there may be a chance to lose some good solutions, which results in the permute convergence of the algorithm.

It is possible that sometimes in the early population, GA finds an optimal solution but gets missed by crossover and mutation rate. In every population, one finest solution is selected and remembered. The elitism technique is used to record this best solution, which is then forwarded to the next generation. Different techniques exist to merge the population to generate a new population, here we use the tournament-based selection method to make a new parent from the existing population. Different parameters used in GA are shown in Table 3. The Algorithm used in the SS is given in Algorithm 1.

---

**Algorithm 1** GA Algorithm used in the SS

---

 1: Initial generation h = 0
 2: Randomly create an initial population representing the appliance patterns
 3: Check the termination criteria, i.e., the maximum generation
 4: Evaluate the fitness of each individual in the population
 5: Select the patterns from the population with the best fitness values; these patterns should represent the chromosomal configuration, which represents the solution
 6: Check the on/off status of all the appliances in the chromosomal configuration
 7: Repeat steps 1–6 for k = 1 as the population size
 8: Select an individual based on fitness and perform mutation
 9: If Pm > Rand, then select the next generation
10: Select two individuals based on fitness and perform crossover
11: If Pc > rand, then crossover this pair
12: Create a new population from the off-springs in 9 and 10
13: h = h + 1, go to step 4 and repeat until h = 24

---

**Table 3.** Control parameters of GA Algorithm.

| Parameters | Values |
|---|---|
| Population size | 100 |
| Maximum generation | 300 |
| Crossover rate | 0.9 |
| Mutation rate | 0.1 |

### 4.2. Binary Particle Swarm Optimization Algorithm

Solution for the same problem, energy consumption patterns, minimization of cost, while considering user comfort and PAR is simulated using BPSO [48]. The SS of HEMC based on BPSO operates at the beginning of the day. After sending a request from the appliance to the SS controller, the action taken by the SS is used to schedule all the appliance's energy consumption patterns in advance. Particle Swarm Optimization (PSO) is one of the best replacements for liner and dynamic programming techniques used to solve complex optimization problems, which is inspired by bird flocking and was developed by Kennedy and Eberhart. The working of PSO is based on the food-searching technique of a swarm of birds in a particular search space. In the search space, two important parameters for each bird are noted, the previous position and velocity. Every bird in the group updates his new position and velocity for the previous position and velocity and knows the position and velocity of the nearest bird to the food court. PSO obtains the same behavior and properties from the bird grouping scenario, and places each particle as a bird which is considered a candidate solution in the search space domain.

The total number of particles in the search space is entitled the population or swarm size. Each particle in the solution search space is studied for its velocity, previous and current position, and fitness value, which represent a solution. To find the finest fitness value for the objective function, BPSO is used having a binary value for optimizing the solution. Each particle in the solution search space represents candidate solutions and obtains the optimal solution each particle has to move in the d-dimensional solution search space.

The preliminary position and velocity of each particle are initialized randomly. To form a swarm, N number of particles are combined. After making a swarm, the particles move around the solution space to obtain the optimal solution. At the end of simulations, the overall best solution called the "global best" is taken as the problem solution. The fitness value for each particle is assessed, and if needed, the local and global positions are updated, respectively. After evaluating the fitness values, every particle in the search space flies and dynamically updates its position and velocity by tracking two extremes, i.e., Plbest and Pgbest in each iteration. The control parameters for BPSO are given in Table 4.

**Table 4.** Control parameters of the BPSO algorithm.

| Parameters | Values |
|---|---|
| Swarm size | 10 |
| Maximum velocity | 3 |
| No. of iteration | 600 |
| C1 | 2.0 |
| C2 | 2.0 |
| Wi | 1.5 |
| Wf | 0.5 |
| Minimum velocity | -3 |

Algorithm 2 shows the BPSO algorithm used in SS. The steps involved in the BPSO algorithm are:

| **Algorithm 2** BPSO Algorithm used in SS |
|---|
| 1: Initialize all the parameters, such as swarm size, no. of iteration |
| 2: Randomly initialize particle (p) for their position and velocity |
| 3: Evaluate fitness of objective function for particle (p) |
| 4: Two best positions global best and local best will be obtained |
| 5: The velocity of each particle is updated, using the inertia weight C1 and C2 |
| 6: The position of each particle is updated |
| 7: Evaluate the fitness of each particle to obtain global best and local best |
| 8: Compare the previous best with the current best |
| 9: Update the global best |
| 10: Repeat until the termination criteria meet |

### 4.3. Wind Driven Optimization Algorithm

Researchers get inspired by nature to solve complex scientific problems in every field of life. The WDO algorithm is one of the nature-inspired algorithms used to solve optimization problems based on atmospheric motion. WDO is an iterative heuristic global optimization algorithm based on population to cope with multi-dimensional and multi-modal problems, having the aptitude to implement different types of constraints on the search domain, as compared to its counterpart GA and BPSO. In principle, very small and tiny particles of air move in an n-dimensional domain, following the second law of motion also used to describe air particle motion within the earth's atmosphere. One prominent factor of WDO, as compared to its other counterpart heuristic algorithm, is to carry out some additional information for velocity updates, such as gravitational force constant and Coriolis forces to give a global best position of the particle with more freedom and robustness. The control parameters for WDO are given in Table 5. Table 6 is used for appliances, consumption power, and LoT, and Table 7 is for the ToU pricing signal.

**Table 5.** Control parameters of the WDO algorithm.

| Parameters | Values |
|---|---|
| Swarm size | 10 |
| Maximum velocity | 5 |
| No. of iteration | 500 |
| C1 | 3.0 |
| C2 | 3.0 |
| Wi | 0.5 |
| Wf | 0.5 |
| Minimum velocity | $-5$ |

**Table 6.** Appliances with power rating and LoT.

| Device | Power Rating (KWh) | LoT (Hour) |
|---|---|---|
| Refrigerator | 0.73 | 21 h |
| TV | 0.50 | 14 h |
| Lighting | 0.6 | 21 h |
| Heater | 4.45 | 3 h |
| Fan | 0.75 | 20 h |
| Iron | 1.5 | 3 h |
| Toaster | 0.05 | 2 h |
| Dishwasher | 3.63 | 3 h |
| Washing machine | 0.78 | 2 h |
| Cloth dryer | 4.40 | 2 h |

**Table 7.** ToU pricing signal.

| High-Peak Hour | Low-Peak Hour | Off-Peak Hour |
|---|---|---|
| 1 am–5 am, 7 pm–10 pm | 5 am–03 pm | 2 am–6 am |

Algorithm 3 shows the WDO algorithm used in SS. The steps involved in the WDO algorithm are:

---
**Algorithm 3** WDO Algorithm used in SS

---
1: Initialized different parameters such as swarm size, pop size, no. of iteration, different coefficient
2: Repeat from h = 0 to h = 24
3: Randomly generates the population of particles
4: Randomly assign velocity and position to each particle
5: Evaluate the fitness of each particle
6: Obtain the local best and global best value for the particles
7: Update the velocity and position of the particle, using inertia and gravitational constant
8: Create a new population
9: Evaluate the fitness of each particle after updating the velocity and position
10: Compare the previous best with the current best particle
11: Update the global best
12: Continue until the termination condition meets

---

## 5. Simulation Results and Discussion

In this section, we are going to discuss the simulation results and graphs for the justification of the proposed HEMC, implemented through Wind Driven Optimization WDO, BPSO, and GA using the ToU pricing scheme. The whole scenario and the proposed model are implemented in the Matlab simulation tool by using the parameters mentioned in Tables 3–7. This research work focuses on calculating energy consumption patterns and electricity costs for different types of appliances with HEMC, without HEMC, and with HEMC using RES, Peak to PAR comparison for different algorithms using schedule load and user comfort while considering appliance waiting time and electricity cost. For our proposed algorithm, we consider different types of appliances having variable length power consumption requirements. HEMC takes the price signal from the utility grid directly through the smart meter and schedules the scheduler according to price signals. Using a two-way communication model, HEMC sends an optimized energy schedule to all the appliances, considering the appliance's consumption patterns and user comfort. The appliances' energy consumption data is taken from a literature review based on reliable data. In our proposed solution, we take 10 different appliances (shiftable, unshiftable, and semi-shiftable) with different energy consumption rates and Lengths of Time (LoT). The attribute: appliance's power rating, number of appliances, and price scheme value are hard coded. For this research paper, an assumption is supposed that household photovoltaic generation must be greater or equal to 35% of its load demand. A time horizon of T = 24 h is considered, which helps the end-user to calculate his/her electricity bill while keeping the constraints of user comfort.

Figure 2 shows the ToU price signal over the 24-h horizon. In the proposed ToU pricing scheme, 24 h of the day are divided into equal intervals. In the ToU pricing model, prices are mostly fixed for a month or season. Based on different incentives and subsidies, the different pricing zone encourages the end-user to schedule and reschedule their daily electricity load to minimize the electricity bill.

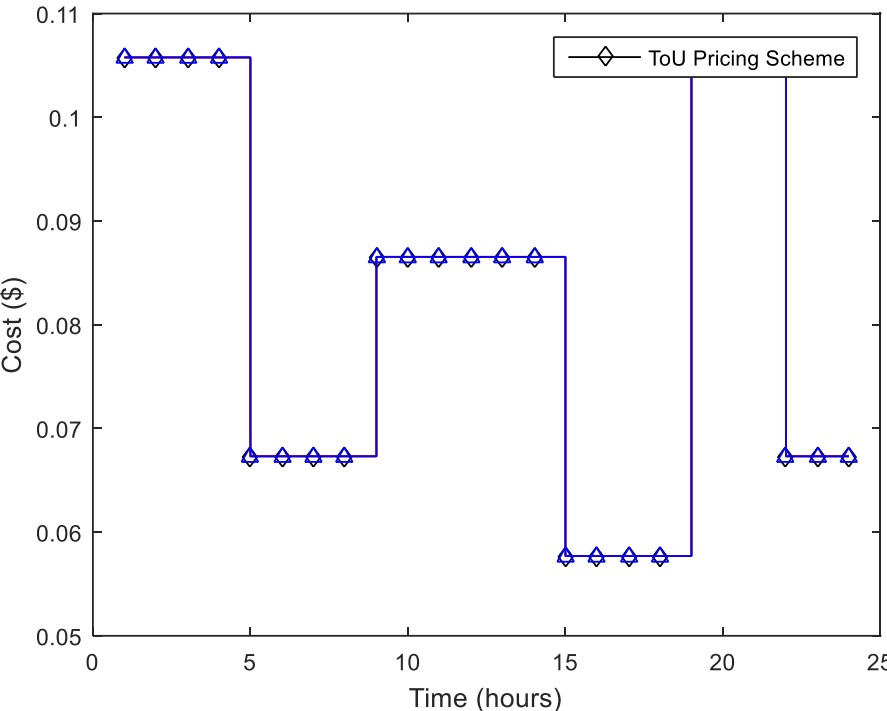

**Figure 2.** ToU pricing signal.

Figure 2 shows different price ranges in 24-h intervals by giving information to the end-user on how to operate his/her appliances to reduce electricity bills. For example, from (0 to 5) h, (9 to 15) h, and (19 to 22) h. In such high-peak-hour intervals, the end-user tries to operate those appliances which consume minimum energy. Similarly, the customer will turn on his maximum appliances in the interval of (5 to 10) h, (15 to 19) h, and (22 to 24) h, where the electricity prices are minimum. In a more practical environment due to the variable energy demand and supply, electricity price also varies which is the core of Demand Response (DR) programs. Thus, using the TOU pricing scheme, if the end-user wants to obtain maximum benefit by reducing the electricity bill he/she must reschedule their load according to the different pricing peaks.

Figure 3 demonstrates the energy consumption pattern for scheduled and unscheduled loads, from the graph it is clear that the smart scheduler schedules the appliances to operate most of the appliances in the shoulder and low price zones, during the time interval (0 to 5), (10 to 15), and (19 to 23) h, the energy consumption of appliances using WDO and BPSO is low as compared to unscheduled and GA. Moreover, the graph shows some peaks in terms of energy consumption where the price is low, in the time interval (5 to 10) and (15 to 19) h, even at the peak the electricity cost will be minimal depending upon the number of appliances operated at that time based on price signal maximizing the user comfort. On the other hand, an unscheduled load does not consider the price signal and all the appliances operate without any schedule. GA shows some strange behavior at the start of the schedule, even in a high-price zone GA turns on more appliances that consume more energy. In the remaining interval, GA almost shows the best results as compared to both WDO and BPSO, on the other hand, BPSO is more prominent than WDO. From the graph, we can conclude that WDO and BPSO more intelligently operate the appliance, by compromising user comfort and more effectively reducing energy consumption and maximizing PAR in the high-price zone. However, overall, the GA result is a little bit more prominent than WDO and BPSO, as from the graph it is clear that in the interval (5 to 24) h, energy consumption is below 10 kw/h for GA. Figure 4 highlights the cost comparison of electricity using WDO, BPSO, and GA algorithms for the appliances using HEMC and without HEMC, respectively. Figure 3 highlights that WDO and BPSO algorithms more intelligently react in low and mid-peak hour intervals by efficiently scheduling the different

home-based appliances, but overall GA gives the best rest as compared to the other two. Since electricity cost is directly related to energy consumption for the same pricing signal, it is clear from the graph that HEMC in low-peak time intervals minimize the energy consumption of different home-based appliances by scheduling them intelligently.

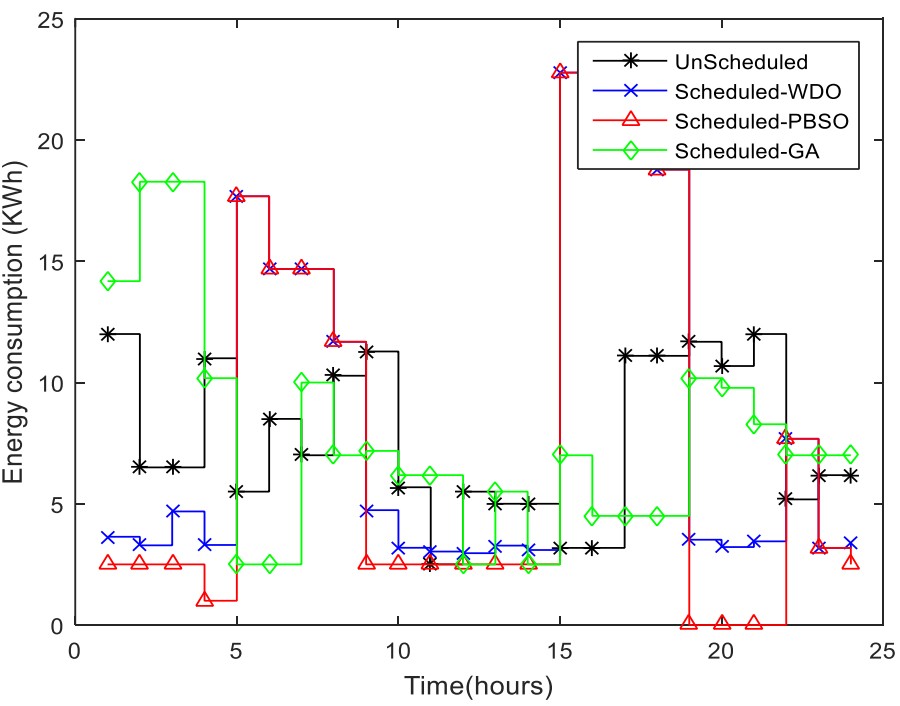

**Figure 3.** Energy consumption comparison of unscheduled and scheduled using WDO, BPSO, GA.

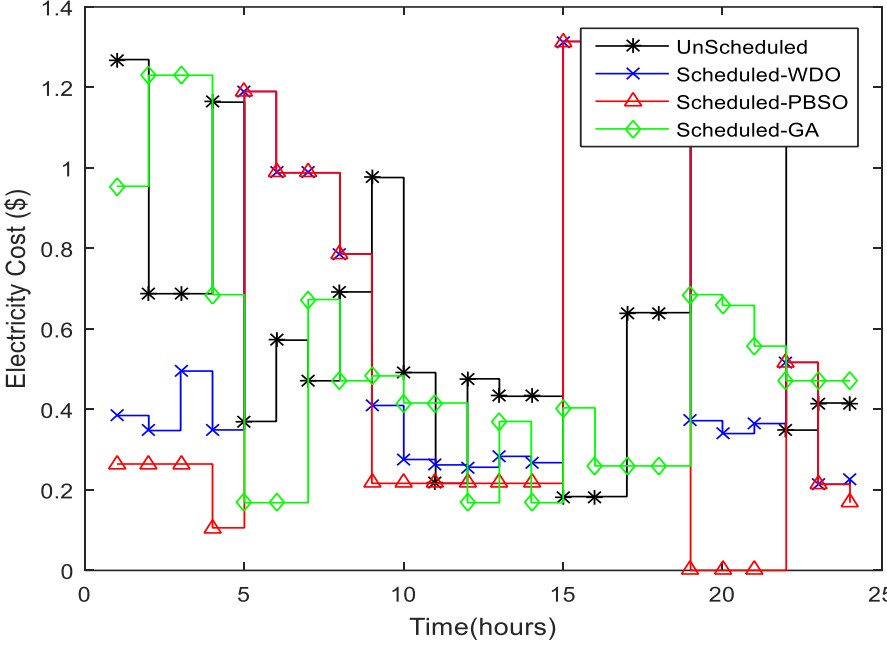

**Figure 4.** Electricity cost comparison of unscheduled and scheduled using WDO, BPSO, GA.

During hours (0 to 4), the scheduled electricity costs of the HEMS system using WDO and BPSO algorithms are comparatively less than unscheduled and GA because HEMC pays the least attention to the maximum capacity bounds and schedules the different appliances by considering the low pricing time interval. The cost of electricity for the WDO and BPSO algorithms is maximum during the high-price time slot (5 to 10), as compared to

unscheduled because HEMC attempts to reschedule a large number of different appliances during these time intervals.

A large number of appliances will be operated in mid-peak hour slots (10 to 15) by GA as compared to WDO and BPSO, resulting in maximum electricity cost. The rest of the cycles for different appliances to be operated are completed during low-price hours (15 to 19). If we compare the cost in discrete quantity: Scheduled with WDO Cost/Day = 12.0748, Scheduled with BPSO Cost/Day = 13.2152, Scheduled with GA Cost/Day = 11.9606, Unscheduled Cost/Day = 15.401. It is clear from the calculations that the electricity bill is higher for unscheduled as compared to WDO, BPSO, and GA, while GA shows better results than WDO and BPSO. If we look at WSO and BPSO, BPSO greatly reduces energy consumption and PAR as compared to WSO. To summarize the discussion, we can conclude that by using the GA algorithm, the electricity cost is effectively reduced, while BPSO and WDO are slightly better than GA in reducing PAR by helping with grid stability and also considering user benefits in terms of electricity bills and comfort levels.

In this scenario, the results of Figures 5 and 6 are compared using WDO, BPSO, and GA optimization techniques for different electrical appliances scheduling by incorporating RES during high-peak hours for smooth and stable functionality of the grid. Cost reduction, peak-to-average reduction, and user comfort are also discussed here. The user's home is equipped with an SS as well as RES and a storage system to store energy. From RES, the user generates and consumes a maximum of 35% of the energy of the daily energy requirement. To reduce energy consumption and minimize electricity bills, appliances considered for this scenario such as washing machines, cloth dryers, iron, and electric vehicle are scheduled in a 24-h time horizon; however, to maximize the comfort level of the end-user, the waiting time for the appliance is also considered, as each appliance has a certain fixed type of interval having a start and finish point. The performance of WDO, BPSO, and GA algorithms is to efficiently consume grid energy as well as the RES is shown in Figure 5.

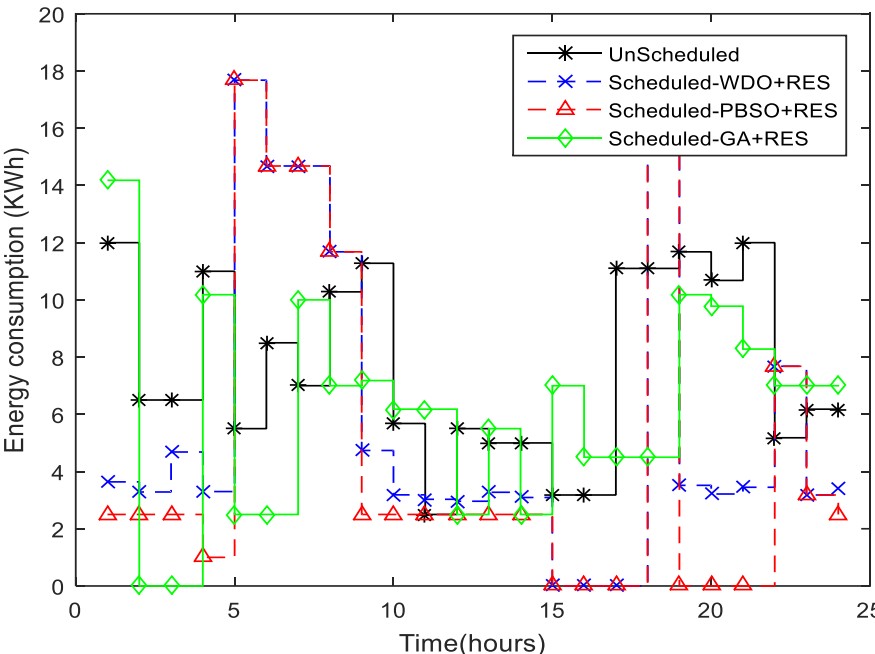

**Figure 5.** Energy consumption comparison of unscheduled, and scheduled with RES using WDO, BPSO, GA.

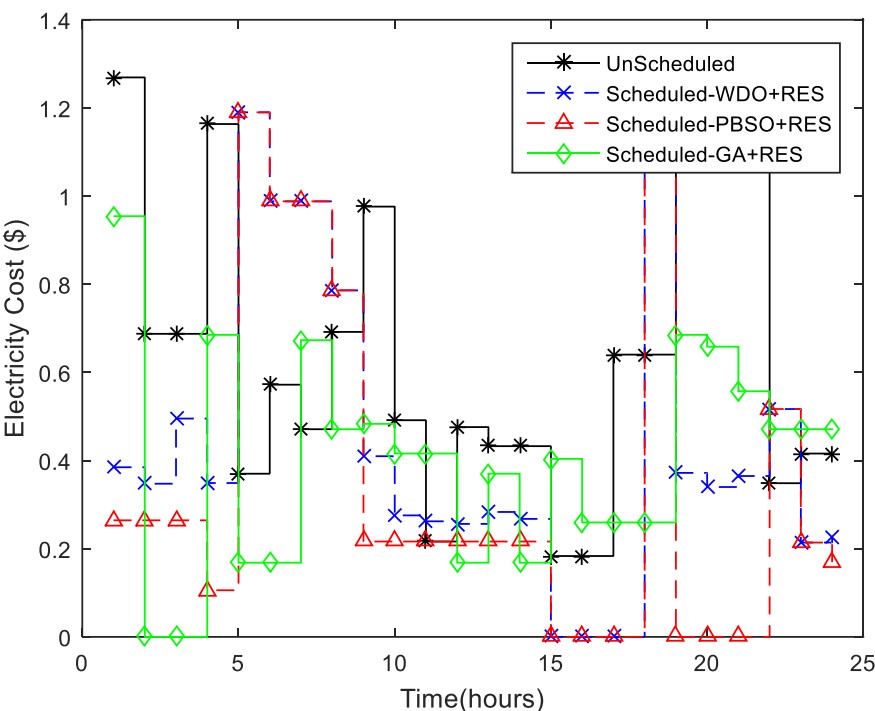

**Figure 6.** Electricity cost comparison of scheduled with RES, scheduled without RES, and Scheduling using WDO.

To minimize the cost of electricity, the Smart Scheduler (SS) employs the RES stored energy, by shifting the load from the grid to RES energy and maximizing user comfort by a very significant amount. The graph demonstrates the minimum energy consumption of different appliances using the smart scheduler having on-site renewable energy in the time interval (0 to 4), and (9 to 21) h for WDO, GA, and BPSO. From Figure 2, it is clear that the price signal is very low at the time interval (15 to 19) h. Using WDO and BPSO algorithms, the smart scheduler schedules the appliance in such a way that in the same interval, the energy consumption becomes 0 by reducing the electricity bill and maximizing the user comfort, as the appliance waiting time here becomes zero, by shifting the appliance from the grid to RES. On the other hand, GA shows some perks at the start interval, for the rest of the interval, GA shows more stable results than WDO and BPSO, while BPSO is more prominent than WDO. The cost of energy of the SS with Photovoltaic (PV) generation is shown in Figure 6. Figure 5, highlights that the SS optimally and efficiently schedules different appliances in low-price signal intervals by shifting the maximum possible load to the RES storage system during the high-peak hour intervals. Using this strategy, the end-user takes maximum benefit from the RES stored energy during high-peak costs and tries to minimize the high peaks in electricity bills. The cost comparison result using HEMC with RES for WDO, BPSO, and GA, without HEMC is given in Figure 6 which highlights the per-day cost reduction in HEMC using RES is maximum as compared to scheduling without HEMC, even in the interval (14 to 19), the cost is almost 0 by maximizing the user comfort.

Figure 7 demonstrates the peak-to-average ratio of different algorithms, i.e., the Genetic Algorithm, Binary Particle Swarm Optimization, and WDO. In a particular time interval, dividing the maximum energy consumption by the average energy consumption of a PAR in a particular time interval for a single user can be calculated. We start to discuss PAR reduction from the very first graph by concluding that generally, end-users want to minimize the total bill of electricity by somehow compromising comfort, while the utility company tries to make sure the stability of the grid in terms of a balanced energy supply. Figure 7 demonstrates the proposed algorithmic technique efficiency by balancing the

energy consumption and minimizing total PAR while paying attention to the capacity constraint of total energy. Figure 7 also shows that WDO and BPSO algorithm remarkably condenses the PAR by 11.48% and 12.48%, respectively, for efficient and optimal home appliance scheduling in low-peak price hours without creating a bottleneck, as compared to the earlier GA algorithm minimizes the PAR by 9.16%, for the same parameter, i.e., many appliances, price signal and power rating for each appliance.

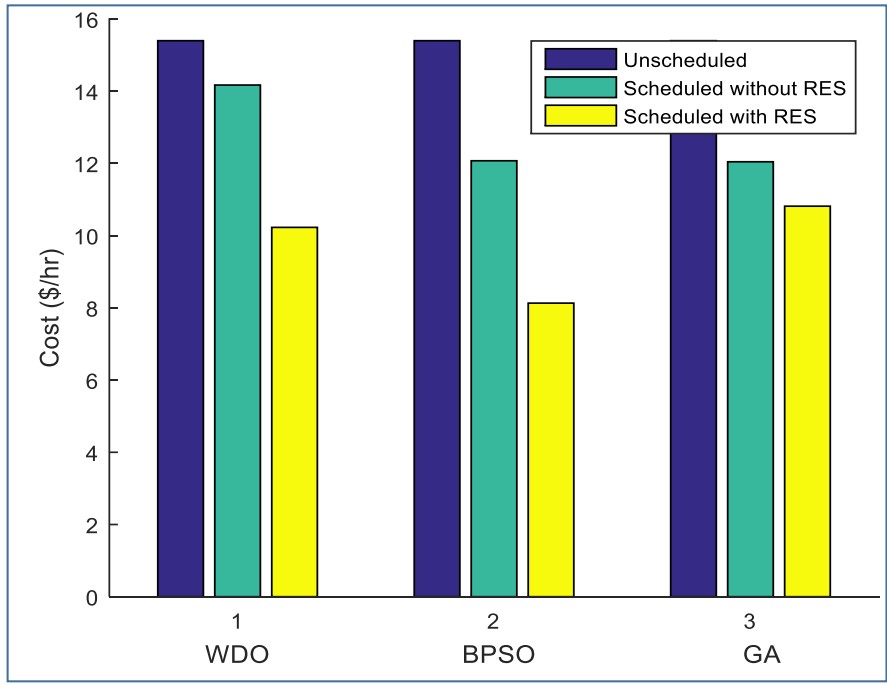

**Figure 7.** PAR comparison of WDO, BPSO, and GA for a scheduled load.

## 6. Conclusions and Future Work

This paper proposed and implemented a home energy management system based on GA, WDO, and BPSO algorithms, for different types of appliances, having different power consumption and length of operation time. Our proposed system uses both the grid and on-site RES energy, optimally scheduling all the different appliances by finding energy consumption patterns, and reducing peak demand and electricity costs by maintaining the user comfort level. From the graph, it is clear that HEMC using GA shows better results than WDO and BPSO, in energy consumption and electricity cost, while BPSO is more prominent than WDO and GA by calculating PAR.

The WDO and BPSO reduce the PAR by 11.48% and 12.48%, respectively, in low-peak price hours without creating a bottleneck, as compared to the earlier GA algorithm which minimizes the PAR by 9.16% for the same parameter. The cost comparison result using HEMC with RES for WDO, BPSO, and GA, without HEMC is given in Figure 6 which highlights that the per-day cost reduction in HEMC using RES is maximum as compared to scheduling without HEMC, even in the interval (14 to 19) the cost is almost 0 by maximizing the user comfort. The proposed HEMC system will disturb performance if the number of appliances is increased. In the future, the number of appliances can be increased by maintaining the user comfort level and the electricity cost. In the future, the proposed model will be upgraded to deploy in multiple locations with different desired flexibility of demand response, connected with various sustainable sources. The relevant techno-economic analysis will be conducted in future research.

**Author Contributions:** This work is carried out in collaboration between all authors. Z.M. and N.A.B. designed a detailed methodology. G.U.R. and M.A. managed the literature searches and detailed analysis. M.Z. wrote the first draft of the manuscript in consultation with G.U.R. The final draft of the paper has been done by A.B. along with M.Z. The funding acquisition is done by B.C. All authors have read and agreed to the published version of the manuscript.

**Funding:** The authors would like to thanks Jiangsu Key Lab of IoT Application Technology, Wuxi Taihu University, Wuxi, China for supporting this work.

**Data Availability Statement:** This article has no associated data.

**Conflicts of Interest:** The authors declare no conflict of interest.

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
