# Peer review of "Efficient Scheduling of Home Energy Management Controller (HEMC) Using Heuristic Optimization Techniques"

_sustainability, doi:10.3390/su15021378_

Round 1
Reviewer 1 Report
The authors propose and implement a home energy management system based on metaheuristic methods for different types of appliances, having different power consumption and length of operation time. The paper is well-written and presents interesting results. However, the authors are requested to address and revise it according to the suggestions listed below:
1. Though the motivating scenarios are clear the contribution of presenting a novel problem is not convincing. The abstract does not clearly explain the contribution.
2. The abstract should be more straightforward for the reader regarding the proposed method and its motivation. The abstract should present some main points for the readers, such as the main contributions, the proposed method, the main problem, the obtained results, the data sets, the comparative methods, etc.
3. The paper lacks the running environment, including software and hardware. The analysis and configurations of experiments should be presented in detail for reproducibility. It is convenient for other researchers to redo your experiments and this makes your work easy acceptance. Add further details on how simulations were conducted. Similarly, system and resource characteristics could be added to Tables for clarity.
4. The discussion of the results needs to include the strengths and weaknesses of the proposed algorithm. The authors should clarify the pros and cons of the methods. What are the limitation(s) methodology(ies) adopted in this work? Please indicate practical advantages, and discuss research limitations. These limitations can be organized around simple distinctions of the choices you made in your study regarding who, what, where, when, why, and how. To have an unbiased view in the paper, there should be some discussions on the limitations of the proposed method.
5. There is not any justification on selecting only three special metaheuristic methods (Genetic Algorithm, Particle Swarm Optimization, and Wind Driven Optimization) for the focused problem.
6. There is not a clear categorization of related work on metaheuristic algorithms. The metaheuristic evolutionary methods are simplistic for this topic. These methods are categorized into nine different classes according to the paper entitled “Plant intelligence based metaheuristic optimization algorithms”. This paper may be considered by citing for brief categorization of metaheuristic algorithms.
7. Statistical tests (such as non-parametric statistical analysis) to judge about the significance of the method’s results is absent. Without such a statistical test, the conclusion cannot be supported. It will be good to present a statistical test in the comparison of the results with other published methods. This can help to support the claim on improved results obtained with the selection methods studied.
8. The robustness about the method has not been discussed. Parameter or sensitivity analysis has not been performed.
9. Are the simulation result taken from the equal conditions? There is not any discussion.
10. Encoding type (representation scheme), fitness function, binarization scheme etc. are not described. These are necessary for the description and adaptation of the metaheuristic methods.
11. Some mathematical notations are not rigorous enough to correctly understand the contents of the paper. The authors are requested to recheck all the definition of variables and further clarify these equations. Definitions of all variables and their intervals should be given.
12. Some paragraphs are too long to read. The authors should try for readability and comprehensibility by dividing paragraphs into two or more.
13. Some more recommendations and conclusions should be discussed about the paper considering the experimental results. The Conclusion section is weak. Furthermore, there is not any discussion section about the results. The conclusion section needs significant revisions. It should briefly describe the findings of the study and some more directions for further research. The authors should describe academic implications, major findings, shortcomings, and directions for future research in the conclusion section. The conclusion in its current for is confused in general. Concerning Conclusion section, it would be better "Conclusions and Future Research", and it is strongly suggested to include future research of this manuscript. What will be happen next? What we supposed to expect from the future papers? So rewrite it and consider the following comments:
- Highlight your analysis and reflect only the important points for the whole paper.
- Mention the benefits
- Mention the implication in the last of this section.
Author Response
Dear Editor,
Please find the attached document of Response to Reviewer 1 Comments.
Regards,
Dr. Ghani Ur Rehman

Reviewer 2 Report
Dear Authors,
I have attached the review report. Please examine it.
Best regards,

Author Response
ear Editor,
Please find the attached document of Response to Reviewer 2 Comments.
Regards,
Dr. Ghani Ur Rehman

Reviewer 3 Report
See attached file

Author Response
ear Editor,
Please find the attached document of Response to Reviewer 3 Comments.
Regards,
Dr. Ghani Ur Rehman

Reviewer 4 Report
This paper proposed and implemented a home energy management system based on GA, WDO, and BPSO algorithms, for different types of appliances, having different power consumption and length of operation time. The subject is very interesting and timely and the paper has been organized well. However, to improve this work, a few modifications need to be implemented:
1- The authors mentioned 5 contributions for this work. Some of them are highlights. I suggest to mention just 1-2 main contributions and afterwards the special features of the work can be mentioned.
2- Since the authors used the heuristic approach, they can mention that the model is not convex and mathematical approaches do not work for the problem so that the readers find out the necessity of using heuristics.
3- From the optimization perspective, how have the parameters been tuned? Please clarify more in the simulation section.
4- The following paper can be suggested for improving the literature review:
For genetic algorithm:
* Salehizadeh, Mohammad Reza, Mahdi Amidi Koohbijari, Hassan Nouri, Akın TaÅŸcıkaraoÄŸlu, Ozan Erdinç, and Joao PS Catalao. "Bi-objective optimization model for optimal placement of thyristor-controlled series compensator devices." Energies 12, no. 13 (2019): 2601.
For demand response:
* Karimianfard, H., Salehizadeh, M. R., & Siano, P. (2022). Economic profit enhancement of demand response aggregator through large scale energy storage system investment. CSEE Journal of Power and Energy Systems.
5- To increase the change of the paper for attracting more readers, I suggest to bring the codes in an appendix or attach them to the paper.
6- please include more numerical findings in the conclusion.
I hope my comments help the authors to improve their good work.
Author Response
ear Editor,
Please find the attached document of Response to Reviewer 4 Comments.
Regards,
Dr. Ghani Ur Rehman

Round 2
Reviewer 1 Report
The current version of the paper presents an expressive improvement as compared to the previous one. The authors provided acceptable answers to all questions and no more issues were detected in the current manuscript. Therefore, this reviewer recommends the acceptance of the paper in its current form.